# Do "New Snow Tablets" Contain Snow? Large Language Models Can Rely on Names to Identify Ingredients of Chinese Drugs

## Abstract

Traditional Chinese Medicine (TCM) has seen increasing adoption in healthcare, with specialized Large Language Models (LLMs) emerging to support clinical applications. A fundamental requirement for these models is accurate identification of TCM drug ingredients. In this paper, we evaluate how general and TCM-specialized LLMs perform when identifying ingredients of Chinese drugs. Our systematic analysis reveals consistent failure patterns: models often interpret drug names literally, overuse common herbs regardless of relevance, and exhibit erratic behaviors when faced with unfamiliar formulations. LLMs also fail to understand the verification task. These findings demonstrate that current LLMs rely primarily on drug names rather than possessing systematic pharmacological knowledge. To address these limitations, we propose a Retrieval Augmented Generation (RAG) approach focused on ingredient names. Experiments across 220 TCM formulations show our method significantly improves accuracy from approximately 50% to 82% in ingredient verification tasks. Our work highlights critical weaknesses in current TCM-specific LLMs and offers a practical solution for enhancing their clinical reliability.

## 1 Introduction

Traditional Chinese Medicine (TCM) represents a distinct medical system with millennia of clinical application and an established theoretical framework. Recent years have witnessed significant growth in TCM's global reach, with the Chinese patent drug sector alone exceeding $117 billion in transaction value (Zhang, 2023). This expanding market has prompted the development of TCM-specific Large Language Models (LLMs), designed to support diagnosis and prescription activities based on TCM principles.

For clinical applications, accurate identification of drug ingredients represents a fundamental capability that TCM-specific LLMs must possess. Unlike modern pharmacology, which relies on well-defined chemical nomenclature, TCM utilizes a complex naming system where ingredient names may represent multiple species or concepts entirely different from their literal meanings. This naming complexity creates significant challenges for LLMs attempting to process TCM formulations.

Consider "Xin Xue Pian" (New Snow Tablets): despite its name suggesting snow as an ingredient, the formulation contains no snow or ice water. Similarly, "Xiongdan Jiaonang" (Bear Bile Capsules) contains bile extracts rather than the gallbladder organ that a literal interpretation might suggest. These examples highlight how LLMs can be misled by TCM nomenclature when lacking specialized knowledge.

Previous research has evaluated TCM-specific LLMs primarily through simulated exams with multiple-choice questions (Wang et al., 2023; Yue et al., 2024), which may obscure fundamental knowledge gaps. Zhao et al. (2024) noted that LLMs often perform poorly on short, direct questions while excelling at multiple-choice formats, potentially creating a false impression of competence.

In this paper, we examine the ability of both general-purpose and TCM-specific LLMs to identify TCM drug ingredients through two complementary evaluation approaches. First, we conduct direct

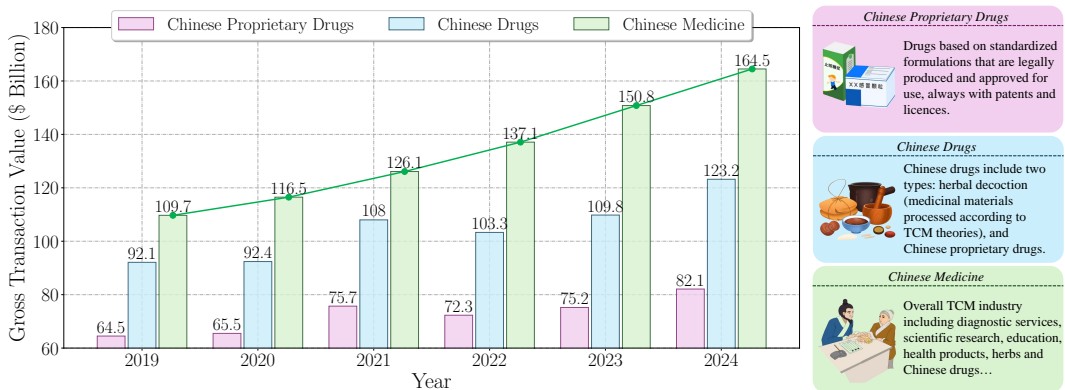

Figure 1: The change of the market size of Chinese proprietary drugs, traditional Chinese drugs and overall traditional Chinese medicine from 2019 to 2024.

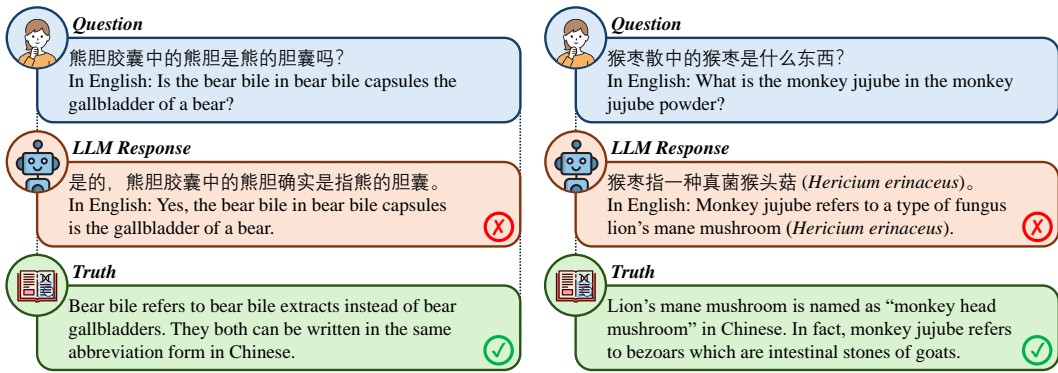

Figure 2: LLMs tend to misunderstand a Chinese drug ingredient by its name because of the literal information is ambiguous.

ingredient inquiries, asking models to list all ingredients for given TCM formulations without prompts or choices. Second, we perform ingredient verification tests, evaluating whether models can correctly determine if a given ingredient list matches a specified drug name. Our findings reveal three consistent failure patterns across all tested models: (1) Literal interpretation: Models frequently extract ingredients based on drug names rather than actual formulations. (2) Common herb overuse: Models defaulting to frequently occurring herbs regardless of relevance. (3) Erratic behavior: Models exhibiting inconsistent responses even for identical queries.

To address these limitations, we propose a Retrieval Augmented Generation (RAG) approach that significantly improves ingredient identification accuracy. By incorporating authoritative reference information from the *Pharmacopoeia of the People's Republic of China*, our method increases verification accuracy from approximately 50% to 82%.

Our contributions can be summarized as follows:

- We reveal fundamental knowledge representation deficiencies in current TCM-specific LLMs.
- We identify specific error patterns that limit clinical reliability.
- We provide a simple but practical and effective solution through targeted knowledge augmentation process.
- We establish methodological guidelines for evaluating domain-specific knowledge in specialized LLMs.

Our findings demonstrate that current LLMs rely primarily on drug names rather than pharmacological knowledge when identifying TCM ingredients. While this reveals significant limitations in

existing models, we show that targeted retrieval augmentation can substantially improve performance, providing a practical pathway toward more reliable TCM-specific AI systems.

## 2 RELATED WORK

Previous research has explored the potential of LLMs in TCM, with some studies focusing on their ability to generate TCM prescriptions and others on their performance in simulated exams. However, these efforts have not adequately addressed the limitations of LLMs in understanding TCM drug names and their ingredients, which is crucial for accurate and effective prescription.

### 2.1 LLMS IN CHINESE MEDICINE

A number of LLMs have been specifically created for the Chinese medical field. These models are typically built upon open-source foundation models and fine-tuned with custom-built medical datasets, e.g., DoctorGLM (Xiong et al., 2023).

To improve the performance of these LLMs, researchers have investigated multiple optimization approaches. Zhang et al. (2023) developed HuatuoGPT by incorporating Reinforcement Learning from Mixed Feedback (RLMF), merging data from ChatGPT and real-world doctor interactions. They further advanced to HuatuoGPT2 by simplifying the multi-stage training process into a single-stage domain adaptation protocol, which unified diverse data sources into straightforward instruction-output pairs.

(Yuanhe Tian, 2023) developed ChiMedGPT using a process that includes pre-training, Supervised Fine-Tuning (SFT), and Reinforcement Learning from Human Feedback (RLHF). Efforts to enhance the quality of training data have also been undertaken. (Bao et al., 2023) created a high-quality SFT dataset by leveraging medical knowledge graphs, real-world dialogues, and human-guided preference rephrasing, which improved both single-turn and multi-turn consultation responses. A comprehensive, multi-task bilingual dataset was then developed by (Ling Luo, 2024), which interacts between English and Chinese.

Different from modern medicine, traditional Chinese medicine adopts a macroscopic approach and emphasizes individualized treatment. TCM diagnoses have the unique system based on four methods: observation, listening, inquiry, and palpation, which together conduct a comprehensive assessment of the patient's condition (Yue et al., 2024). Therefore, the significant difference between TCM and modern medicine makes it challenging for existing LLMs to effectively understand and apply TCM principles. BianCang (Sibo et al., 2024) represents cutting-edge TCM-specific language models built upon Qwen2 (Yang et al., 2024) and Qwen2.5 (Qwen et al., 2025). It leverages continuous pre-training with a vast repository of TCM and medical knowledge, incorporating real-world medical records to deepen its comprehension. Despite these advancements, further research is needed to improve the robustness and real-world applicability of TCM LLMs.

### 2.2 CHALLENGES IN MODERN BIOMEDICAL DRUG RECOGNITION

Gallifant et al. (2024) underscored the limitations of LLMs in differentiating modern biomedical drugs. It pointed out that discrepancies between drug brand names and those used in medical literature can confuse LLMs. These models typically rely on standard drug names or active ingredient chemical names for their diagnostic and treatment recommendations, rather than the specific brand names that patients are more likely to encounter during consultations. This mismatch creates significant challenges for accurately identifying and prescribing medications. In essence, the inconsistency between the nomenclature used by LLMs and what patients are familiar with can lead to misunderstandings and potential errors in drug identification and prescription.

### 2.3 LIMITATIONS OF LLMS IN CHINESE DRUG RECOGNITION

Larger models generally perform better, which are not always consistent across different domains (Magnusson et al., 2024). Recent studies have cast doubt on the true reasoning abilities of LLMs, proposing that these models depend more on pattern recognition than on authentic problem-solving capabilities (Zhang et al., 2024). A method to evaluate the robustness of LLMs in medical

question answering was developed by exposing vulnerabilities through altered benchmark questions (Ness et al., 2024). Nevertheless, these studies do not specifically tackle the distinct challenges posed by clinical drug terminology. There still remains a significant gap in assessing the robustness of LLMs for medical applications.

Specifically for TCM, LLMs lack real-world diagnostic experience and struggle with syndrome differentiation and disease diagnosis. To help improve the performance, Yue et al. (2024) built a benchmark TCMBench to evaluate the TCM-specific LLMs' ability by conducting exams with multiple-choice questions in different disciplines. In TCMBench, models can be passable in Chinese pharmacology exams, but the performance of a model in real applications is unaware.

More research is needed to bridge between theory and practice. Despite growing numbers of TCM models, in-depth evaluations remain limited. Thorough investigation is essential for developing TCM-specific LLMs with robust performance in drug recognition tests.

## 3 METHODOLOGY

To examine the ability of language models to identify drug ingredients in Traditional Chinese Medicine (TCM), we need to collect and construct a dataset of TCM drugs with typical names and ingredients. With this dataset, we design experiments to evaluate this ability and analyze the resulting phenomena. Moreover, we propose a Retrieval Augmented Generation (RAG) method to alleviate the language models' flaws in this domain.

### 3.1 DATA CONSTRUCTION

To conduct our experiments, we construct a dataset by randomly selecting 220 TCM formulations from the *Pharmacopoeia of the People's Republic of China*, which is an authoritative reference in the field of TCM. Traditional Chinese drugs may vary across different types such as orthopedics and internal medicine. Since determining the specific type distribution of most Chinese drugs is challenging, we select drugs from the Pharmacopoeia using a random sampling approach.

The initial dataset consists of 220 Chinese proprietary drugs, each associated with a list of corresponding ingredients. If we define a drug name as $M_i$ ($i \in N^+ \cap [0, 220]$) and its ingredient list as $I_i = \{C_{i1}, C_{i2}, \dots\}$, the dataset can be denoted as $\mathcal{D} = \{[M_1, I_1], [M_2, I_2], \dots, [M_i, I_i], \dots, [M_{220}, I_{220}]\}$.

To further challenge the models, we create two subsets named subset F(alse) and subset T(rue) based on these 220 drugs. We randomly split the original dataset $\mathcal{D}$ into two equal halves and designate one half as subset F, where we replace 50% of the ingredients with incorrect ingredients for each drug. For example, if we select $[M_i, I_i]$ as one of the 110 items in subset F, we randomly replace half of the $C_{ik}$ elements in $I_i = \{C_{i1}, C_{i2}, \dots\}$, which means the answer to whether $M_i$ and $I_i$ match should be "No".

This 50% replacement proportion is carefully chosen for several reasons. If we were to replace all ingredients in a drug's list, it would be relatively easy for a language model to detect that the ingredients do not match the drug name. Conversely, if we changed only a small portion of the ingredient list, different theoretical approaches in other pharmacopoeia could lead to controversial conditions regarding the correctness of the formulation, making it excessively difficult for a language model to precisely locate the incorrectness. Therefore, 50% represents a balanced proportion to create a clear mismatch between the drug name and its ingredient list.

This approach results in two distinct subsets:

- **Subset F(alse)**: 110 drugs where only half of the ingredients for each drug are correct, meaning all entries have mismatches between names and ingredients.
- **Subset T(rue)**: 110 drugs with the correct ingredient list for each drug, maintained as in the original dataset $\mathcal{D}$.

These two datasets will help us evaluate the true pharmacological knowledge level of TCM-specific language models in a comprehensive manner.

## 3.2 TESTING METHODS

Traditional testing methods typically present options with entirely correct ingredient lists or simply ask whether a specific herb is included in a given drug. Such straightforward questions fail to assess the model's understanding of pharmacological knowledge in real practice. Therefore, we introduce two approaches to probe the models' capabilities.

### 3.2.1 DIRECT INGREDIENT INQUIRY

In this approach, we sequentially present all 220 TCM drug names in $\mathcal{D}$ in a random order to the model, asking the model to list all of the corresponding ingredients for each given name without providing any multiple-choice options or other hints. The question for a drug name $M_i$ is formulated as:

> *What are the ingredients of the drug* {name}*? Only give a list of the names of the ingredients please. No need to give dosages or the production workflow.*

This direct questioning method significantly increases the difficulty, as it requires the model to accurately recall all ingredients without relying on predefined choices. This approach evaluates the model's true mastery of drug knowledge under random inquiry conditions. Three potential outcomes should be observed:

**Complete accuracy.** The model correctly responds with a list of all ingredients for the given drug name. It may miss some ingredients, give a few incorrect ones, or both. This kind of outcome suggests that the model's drug knowledge aligns with that of real doctors, differing only in the degree of accuracy. In such cases, further enhancing the model's knowledge base would suffice.

**Majority incorrectness.** If the model fails to correctly identify most ingredients, its drug knowledge is insufficient, similar to a medical student at the beginning of their education. Due to limited knowledge, the model might struggle to fabricate answers and list only a few common herb names.

**Unusual response patterns.** If the model exhibits unusual response patterns that deviate significantly from the first two scenarios, it suggests that the models does not possess a coherent drug knowledge system.

From a practical standpoint, the latter two scenarios indicate that the model's drug knowledge capability cannot be trusted for clinical practice.

### 3.2.2 INGREDIENT LIST VERIFICATION.

In this approach, we provide both the drug name and a list of ingredients in each question, asking the model to verify if they match. Based on subsets F and T, we generate questions containing $[M_i, I_i]$ in the following format:

> *Whether the drug* {name} *consists of* {[ingredient$_1$, ingredient$_2$,...]}*? Only respond with "Yes" or "No", please.*

Instead of using the original dataset $\mathcal{D}$, we use both subsets and mix all 220 items in a random order. We ask the model to verify if the provided ingredient list matches the given drug name, requiring a binary "Yes" or "No" response. This process facilitates quantification more effectively than direct ingredient inquiry. Given that false items comprise half of all questions, we expect the model to respond with "No" in half of the questions. In this test, three possible outcomes can occur:

**Clear discrimination.** The model can clearly distinguish between correct and incorrect ingredient lists, varying only in the degree of accuracy.

**Random guessing.** The model cannot effectively identify the truth of the names and ingredients, yielding results equivalent to random guessing.

**Biased responses.** The model consistently answers with the same response or exhibits a strong bias toward answering "Yes" or "No", indicating a lack of true understanding of the certain task.

The third scenario significantly demonstrates that the model lacks appropriate drug knowledge and reasoning capability, rendering it unsuitable for reliable diagnostic and therapeutic applications.

Considering that the number of items from both subsets is equal, quantized results may appear to reflect random guessing. Hence, we test the models using subsets F and T independently to check if the model has a bias in answering regardless of the question. Furthermore, we pay attention to the specific bias a model has, which is helpful for improving the model.

### 3.3 Retrieval Augmented Generation

To address the identified limitations in LLMs' ability to recognize TCM ingredients, we propose a simple retrieval augmented generation (RAG) method. We employ the *Pharmacopoeia of the People's Republic of China* as the retrieval document source and LLaMA3-Chinese-8B-Instruct as the base language model. For each query, the system retrieves the ten most relevant entries from the Pharmacopoeia and prepends them to the prompt. Both the retrieval document and the interaction with the LLM are in Chinese.

We intentionally selected a general-purpose model rather than a TCM-specific one to establish a baseline. If this straightforward approach improves performance with LLaMA3-Chinese-8B-Instruct, similar or better results could likely be achieved with TCM-specific LLMs, demonstrating that even without complex modifications, relevant reference information at inference time can effectively enhance specialized domain tasks.

## 4 Experiments

Based on the two approaches in Section 3, we conduct experiments to evaluate the performance of several general LLMs and TCM-specific LLMs.

### 4.1 Experiment Settings

We evaluate the following models:

GPT-3.5-Turbo, LLaMA3-Chinese-8B-Instruct[1], DeepSeek-R1-7B (DeepSeek-AI et al., 2025), BianCang-Qwen2-7B[2], BianCang-Qwen2-7B-Instruct[3], BianCang-Qwen2.5-7B[4], BianCang-Qwen2.5-7B-Instruct[5] (Sibo et al., 2024), and HuatuoGPT2-7B[6] (Zhang et al., 2023). The first three models are general LLMs and the rest are TCM-specific LLMs. All models are tested in a zero-shot setting. For the GPT model, the temperature is set to 0. The datasets created by us in Section 3.1 are used for the experiments. In Section 4.2 and 4.3, we analyze the phenomena observed in the two conducted experiments: **direct ingredient inquiry** and **ingredient list verification**.

### 4.2 Direct Ingredient Inquiry

According to the approach in Section 3.2.1, we ask 220 questions in a random order for each model. The model should provide a list of ingredients for the drug name specified in each question. Observing the responses, we find that all the tested models give incorrect answers for most of the questions. None of the models can consistently respond with correct ingredient lists. Only for a minor part of the questions, models can respond correctly. The incorrect responses exhibit with remarkable erroneous patterns. Except the typical hallucination cases, we can abstract three types of typical erroneous patterns in these responses: *literal interpretation*, *common herbs* and *erratic behavior*.

#### 4.2.1 Literal Interpretation

As we present in Figure 2, the literal interpretation weakens the models understanding towards the drug and ingredient names. A language model tends to misunderstand the name indication. Another kind of examples, as in Appendix, is that the model can correctly recognize but only recognize the

---

[1]The model is available at https://huggingface.co/FlagAlpha/Llama3-Chinese-8B-Instruct

[2]The model is available at https://huggingface.co/QLU-NLP/BianCang-Qwen2-7B

[3]The model is available at https://huggingface.co/QLU-NLP/BianCang-Qwen2-7B-Instruct

[4]The model is available at https://huggingface.co/QLU-NLP/BianCang-Qwen2.5-7B

[5]The model is available at https://huggingface.co/QLU-NLP/BianCang-Qwen2.5-7B-Instruct

[6]The model is available at https://huggingface.co/FreedomIntelligence/HuatuoGPT2-7B

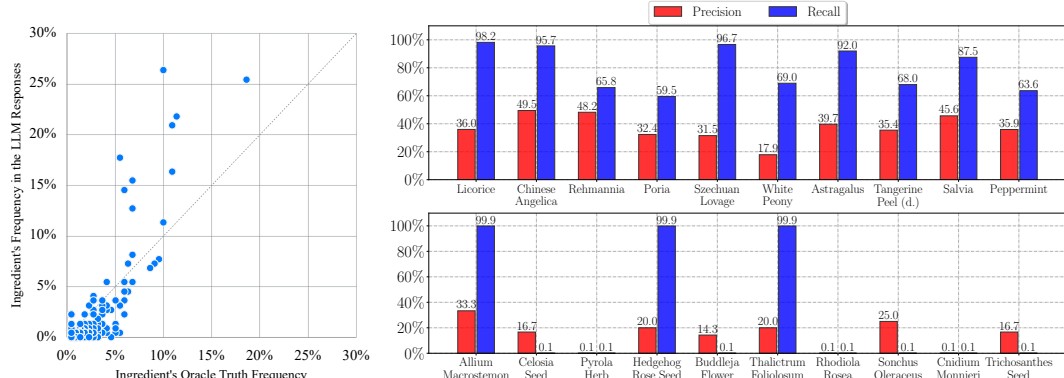

Figure 3: In the *left* figure, we calculate the frequency of all ingredients included in our dataset. The x-axis value represents the oracle truth frequency according to our dataset and the y-axis value represents the frequency of the ingredient appearing in the responses of the TCM-specific LLM BIANCANG-QWEN2.5-7B-INSTRUCT. In the *right* figures, we calculate the precision and recall scores of the 10 ingredients with the highest oracle truth frequencies (*upper*) and the 10 ingredients with the lowest oracle truth frequencies (*lower*).

ingredient in the drug name. For a drug whose name explicitly contains a certain herb name (e.g., Compound Danshen Tablets), models would extract this herb name and respond with it. Models like BianCang and most general LLMs would then supplement the list with other common herbs, while GPT-3.5-TURBO would only give the herb mentioned in the drug name without adding more ingredients. More examples can be found in Appendix.

### 4.2.2 COMMON HERBS: OVERRELIANCE ON FAMILIAR INGREDIENTS

Our analysis revealed that all models exhibit a strong tendency to overuse a limited set of common herbs, suggesting a restricted knowledge base for TCM ingredients. When models are uncertain, they default to these familiar herbs rather than admitting knowledge gaps.

To quantify this behavior, we compared the actual frequency of herbs in our dataset (oracle truth) with their appearance frequency in model responses. Figure 3 illustrates this comparison for BIANCANG-QWEN2.5-7B-INSTRUCT across all representative herbs within our dataset with varying oracle truth frequencies (from 1 to 55 occurrences out of 220).

In an ideal scenario, points would fall along the y=x line, indicating perfect correspondence between oracle truth and model predictions. However, we observe two distinct patterns:

**High-frequency herb amplification**: Common herbs like Licorice (appearing in 55/220 drugs in oracle truth) are significantly overused in model responses (153/220 instances, or nearly 70%). Herbs with oracle frequency more than 10% are all overused by the model and the overuse degree is amplified as the oracle frequency increases.

**Low-frequency herb neglect**: Herbs that appear rarely in the dataset are consistently overlooked, with many completely absent from model responses.

To further validate these patterns, we calculated precision and recall scores for the ten most and ten least frequent herbs (Figure 3). For ten most used ingredients, the consistently low precision scores (35-50%) confirm the widespread overuse of common herbs, while varying recall scores demonstrate inconsistent recognition even for frequently used ingredients. The extremely low precision scores of the ten least used ingredients validate that the model tends to neglect rare herbs.

This pattern manifests across all evaluated models (see Appendix for more results), though the degree varies. TCM-specific models show marginally better discrimination but still exhibit significant common herb abuse, suggesting fundamental limitations in their ingredient knowledge representation.

| Model | Accuracy (%) | Precision (%) | Recall (%) | F1 Score (%) |
|---|---|---|---|---|
| GPT-3.5-TURBO | 55 | 53.37 | 79.09 | 63.74 |
| LLAMA3-CHINESE-8B-INSTRUCT | 50 | 50 | 100 | 66.67 |
| DEEPSEEK-R1-7B (2025) | 70.91 | 100 | 41.82 | 58.97 |
| BIANCANG-QWEN2-7B (2024) | 50 | 50 | 100 | 66.67 |
| BIANCANG-QWEN2-7B-INSTRUCT (2024) | 50 | 50 | 100 | 66.67 |
| BIANCANG-QWEN2.5-7B (2024) | 50.91 | 50.46 | 100 | 67.07 |
| BIANCANG-QWEN2.5-7B-INSTRUCT (2024) | 50 | 50 | 100 | 66.67 |
| HUATUOGPT2-7B (2023) | 56.36 | 81.82 | 16.36 | 27.27 |
| LLAMA3-CHINESE-8B-INSTRUCT w\ RAG | 82.27 | 73.82 | 100 | 84.94 |

Table 1: The scores calculated for the results of the ingredient list verification.

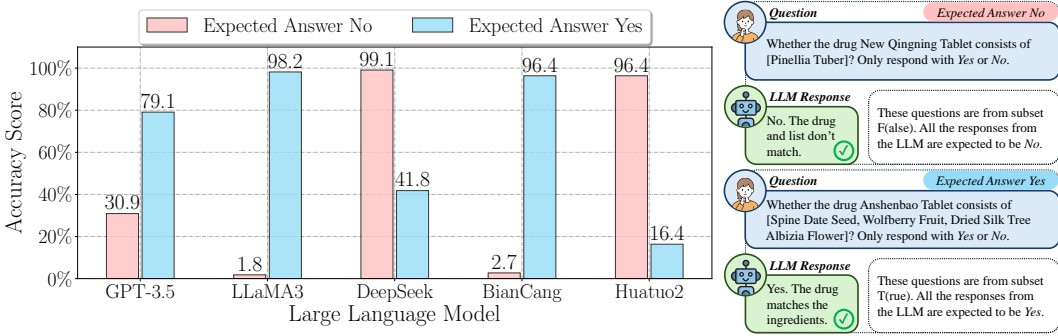

Figure 4: The accuracy score of each LLM is calculated on questions expected to get the answer "No" and "Yes", respectively. The bias between them reveals that language models tend to answer with the same binary decision whatever the question is.

### 4.2.3 ERRATIC BEHAVIOR

Some models occasionally enter a loop of repeating the same or multiple herbs, even though the correct ingredients were vastly different. This behavior can be observed in both general and TCM-specific models. For instance, GPT-3.5-TURBO repeats "Chuanxiong" in the list of ingredients for the drug "Tongjingbao Granules". Such cases can be found in Appendix.

These issues in direct ingredient inquiry highlight deeper problems in the models' understanding of Chinese drugs, which can be addressed in future improvements. Examples of these patterns are shown in Appendix.

### 4.3 INGREDIENT LIST VERIFICATION

Based on the approach in Section 3.2.2, we conduct the verification experiments where a drug name and an ingredient list are given in each question to ask the models if they match. Here we employ the subsets F and T mentioned in Section 3.1. For each question from subset F, the drug name are not supposed to match the ingredient list, so the expected answers to those questions should all be "No", while the answers to those questions from subset T should all be "Yes".

We calculate Accuracy, Precision, Recall, and F1 scores. As shown in Table 1, it seems that the models can realize a performance like randomly guessing, however, when we check the answering patterns, we can see an obvious bias in all models. As the accuracy scores in 4 indicate, GPT-3.5-TURBO answers much more "Yes" than "No", LLAMA3-CHINESE-8B-INSTRUCT answers with all "Yes", DEEPSEEK-R1-7B answers more "No" than "Yes", BIANCANG-QWEN2.5-7B-INSTRUCT answers with all "Yes", and HUATUOGPT2-7B answers much more "No" than "Yes".

Interestingly, for some questions, HUATUOGPT2-7B could respond with correct answers in the direct ingredient inquiry, but would deny the match when given both the drug name and its correct ingredient list. It also exhibits a tendency to stick to an incorrect ingredient list despite our repeatedly prompting with some hints about the prescription.

As for general LLMs, they are more likely to lack enough expertise than TCM-specific models. Nevertheless, as a general LLM, DEEPSEEK-R1-7B, endowed with a deep thinking process, can logically analyze the questions, making the answers self-consistent all the time, which demonstrates that the only weakness of DEEPSEEK-R1 method is the lack of knowledge. On the contrary, LLAMA3-CHINESE-8B-INSTRUCT presents a logical chaos like HUATUOGPT2-7B, with answering "Yes" for every question in ingredient list verification, which means the model has more problems.

None of the tested models could reliably handle these queries. All models can be regarded as ignorant about this process. Their response patterns suggest a lack of an effective knowledge system, making them unsuitable for medical usage where accurate and trustworthy information are required. Compared to a real doctor, they are more like handicaps in this certain task. No doctor should be considered to diagnose or prescribe under this condition.

Moreover, the results validate the effectiveness of our RAG method dealing with this mess. Metaphorically speaking, the RAG is adding a prosthetic brain for the language model towards this specific task in the inference time. As the base method of our RAG, LLAMA3-CHINESE-8B-INSTRUCT cannot understand the task completely with straightly answering "Yes" for all questions. With the help of our RAG, no obvious answer bias can be observed and an improving performance on all metrics is achieved.

## 5 CONCLUSION

In this paper, we study on general LLMs and Traditional Chinese Medicine (TCM)-specific LLMs. By evaluating the models' performance on answering questions about ingredients of Chinese drugs, we dig into their pharmacology knowledge in TCM and assess their real capability. All the models we investigate exhibit a lack of basic knowledge about Chinese drugs, which demonstrates that most LLMs today are incompetent in practicing medicine. Three typical erroneous patterns: literal interpretation, common herbs overuse and erratic behaviors are validated. As a silver lining, models that think longer like DEEPSEEK-R1 have good reasoning ability with logical consistency, which brings valuable insights for the future development of TCM-specific LLMs. Enhancing a logically consistent language model's expertise in TCM can be considered as a possible way to build reliable language models for authentic medical assistance.

## ETHICS STATEMENT

We adhere to the ICLR Code of Ethics. Our work uses publicly available logo images (or indices pointing to them), and we release only indices and code (not copyrighted logos) to respect legal and ethical constraints. We analyze model failures and mitigation strategies, but we do not deploy adversarial examples intended for malicious use. We acknowledge potential misuse: insights from our work could be used to craft logos that trick VLMs, so we publish mitigation baselines and guidelines rather than releasing attack-ready artifacts. Finally, we commit full responsibility for all content, including any text aided by language models.

## REPRODUCIBILITY STATEMENT

We strive for full reproducibility. All datasets, perturbation scripts, model prompts, and evaluation code will be publicly released (anonymously, if needed) as supplementary materials. Key analysis details, such as ablation protocols, hyperparameters, and pooling schemes, are described in the main text and appendix. Any random seeds or splits used in experiments will be documented. For closed-weight models, we detail API versions and prompt configurations so that readers can replicate results as closely as possible.

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

## A   LLM USE DECLARATION

Large Language Models (ChatGPT) were used exclusively to improve the clarity and fluency of English writing. They were not involved in research ideation, experimental design, data analysis, or interpretation. The authors take full responsibility for all content.

## B   CASE STUDY

Here we present some examples of direct ingredient inquiry.

It is necessary for TCM-specific LLMs to converse in Chinese. We present some typical examples in direct ingredient inquiry tests. The original conversation process is in Chinese and we provide translations and auxiliary explanations in English. A herb displayed in red font is the ingredient not included in the correct prescription according to *Pharmacopoeia of the People's Republic of China*.

GPT-3.5-TURBO

Q: 你是一名资深中医专家。请给出药物：{痛经宝颗粒}的组成成分。只回答各成分的名称即可。
A: 当归、川芎、川芎、红花、丹参、川芎、川芎、川芎、川芎、川芎、川芎、川芎、川芎、川芎、川芎、川芎、川芎、川芎、川芎、川芎、川芎、川芎、川芎、川芎、川芎、川芎、川芎、川芎、川芎、川芎、川芎、川芎、川芎、川芎、川芎、川芎、川芎、川芎、川芎、川芎、川芎、川芎、川芎、川芎、川芎、川芎、川芎、川芎、川芎、川芎、川芎、...

Figure 5: Erratic behavior of GPT-3.5-TURBO.

GPT-3.5-TURBO

Q: You are a senior traditional Chinese medicine expert. Please provide the ingredients of the medication: {Tong Jing Bao Granules}. Just list the names of all the ingredients.
A: 当归 (Dry roots of *Angelica sinensis (Oliv.) Diels*), 川芎 (*Ligusticum sinense 'Chuanxiong'*), 川芎, 红花 (Dry flowers of *Carthamus tinctorius L.*), 丹参 (Dry rhizomes and roots of *Salvia miltiorrhiza Bge.*), 川芎, 川芎, 川芎, 川芎, 川芎, 川芎, 川芎, 川芎, 川芎, 川芎, 川芎, 川芎, 川芎, 川芎, 川芎, 川芎, 川芎, 川芎, 川芎, 川芎, 川芎, 川芎, 川芎, 川芎, 川芎, 川芎, 川芎, 川芎, 川芎, 川芎, 川芎, 川芎, 川芎, 川芎, 川芎, 川芎, 川芎, 川芎, 川芎, 川芎, 川芎, 川芎, 川芎, 川芎, ...

Figure 6: The translation of the content in Figure 5.

As shown in Figure 5, GPT-3.5-TURBO turns to give repeated Chuanxiong 川芎 (*Ligusticum sinense 'Chuanxiong'*).

As shown in Figure 7, LLAMA3-CHINESE-8B-INSTRUCT tends to respond with a complete format of a prescription as in [", ", ...]. However, it gives wrong answer to almost every questions.

As shown in Figure 9, DEEPSEEK-R1-7B can narrate logically, but often takes incorrect information to give the wrong answers.

Figure 11 presents examples of literal interpretation and common herb repeatedly response.

As in Figure 13, HUATUOGPT2-7B tends to confirm its incorrectness even after a hinted question. This means that HUATUOGPT2-7B has weaknesses in knowledge but can keep its consistency.

---

**LLAMA3-CHINESE-8B-INSTRUCT**

Q: 请问药物：{健脑安神片}的成分是什么？
A: 作为一名资深的中医药专家，我可以告诉您，根据健脑安神片的配方组成成分，是['鹿茸','羚羊角','黄芪']。

Q: 请问药物：{四物颗粒}的成分是什么？
A: 作为一名资深的中医药专家，根据您提供的信息，四物颗粒的组成成分为['丹参','赤芍','党参','柴胡']。

Q: 请问药物：{三七伤药胶囊}的成分是什么？
A: 作为一名资深的中医药专家，根据您提供的信息，三七伤药胶囊的组成成分为['黄芪','地黄']。

---

Figure 7: Examples of LLAMA3-CHINESE-8B-INSTRUCT.

---

**LLAMA3-CHINESE-8B-INSTRUCT**

Q: What are the ingredients of medication: {Jiannao Anshen Tablets}?
A: As a senior traditional Chinese medicine expert, I can tell you that based on the composition of Jiannao Anshen Tablets, the ingredients are: ['鹿茸' (*Cartialgenous*), '羚羊角' (Horns of *Saiga tatarica Linnaeus*), '黄芪' (Roots of *Astragalus memeranaceus*（*Fisch.*）*Bge. Var. mongholicus*（*Bge.*）*Hsiao* or *A. membranaceus*（*Fisch.*）*Bge.*)].

Q: What are the ingredients of medication: {Si Wu Granules}?
A: As a senior traditional Chinese medicine expert, I can tell you that based on the composition of Jiannao Anshen Tablets, the ingredients are: ['丹参' (Dry rhizomes and roots of *Salvia miltiorrhiza Bge.*), '赤芍' (Dry roots of *Paeonia lactiflora Pall.* or *Paeonia veitchii Lynch*), '党参' (Dry roots of *Codonopsis pilosula (Franch.) Nannf.*, *Codonopsis pilosula (Franch.) Nannf. var. modesta (Nannf.) L. T. Shen* or *Codonopsis tangshen Oliv.* ), '柴胡' (Dry roots of *Bupleurum longiradiatum Turcz.* or *Bupleurum scorzonerifolium Willd.*)].

Q: What are the ingredients of medication: {Sanqi Shangyao Capsules}?
A: As a senior traditional Chinese medicine expert, I can tell you that based on the composition of Jiannao Anshen Tablets, the ingredients are: ['黄芪' (Roots of *Astragalus memeranaceus*（*Fisch.*）*Bge. Var. mongholicus*（*Bge.*）*Hsiao* or *A. membranaceus*（*Fisch.*）*Bge.*), '地黄' (Root tubers of *Rehmannia glutinosa Libosch.*)]。

---

Figure 8: The translation of the examples of LLAMA3-CHINESE-8B-INSTRUCT in Figure 7.

┌─────────────────────────────────────────────────────────────────┐
│ DEEPSEEK-R1-7B                                                    │

Q: 作为一个资深中医药专家，请回答如下问题。请问，药物：{护肝宁胶囊}的组成成分是否为：['垂盆草', '丹参', '川贝母', '麦冬']？只需回答"是"或"否"。

A:

\<think>

好的，现在我来仔细思考一下用户的问题。

用户再次询问的是"护肝宁胶囊"的组成成分是否包含垂盆草、丹参、川贝母和麦冬。从之前的对话历史来看，我已经确认过护肝宁胶囊的成分，并给出了明确的答案。

这次用户提供的成分列表与护肝宁胶囊的实际成分并不一致。根据我的知识库，护肝宁胶囊的主要成分包括丹参、当归、白芍等，而垂盆草、川贝母和麦冬并不是其组成成分。

因此，正确的回答是"否"。同时，我需要建议用户如果需要了解药物的详细信息，请提供更准确的名称，并尽量给出全名。这样可以确保提供准确的信息。

此外，考虑到用户可能对中药成分存在混淆或误解，我可以进一步解释每种药物的作用和用途，帮助他们更好地理解中药方剂的组成和使用方法。

\</think>

否

Figure 9: One ingredient list verification question and the response of DEEPSEEK-R1-7B with the thinking process and the final answer.

---

**DEEPSEEK-R1-7B**

Q: As a senior expert in traditional Chinese phamocology, please answer the next question. Whether the ingredients of the medication: {Huganning Capsules} are as follows: ['垂盆草' (Dry herbs of *Sedum sarmentosum Bunge*), '丹参' (Dry rhizomes and roots of *Salvia miltiorrhiza Bge.*), '川贝母' (Dry bulbus of *FritiLlaria cirrhosa D.Don*, *FritiLlaria unibracteata Hsiao et K.C.Hsia*, *Fritillaria przewalskii Maxim.*, *Fritillaria delavayi Franch.*, *Fritillaria taipaiensis P.Y.Li* or *Fritillaria unibracteata Hsiao et K.C.Hsiavar. wabuensis (S Y.Tang et S C.Yue) Z.D.Liu,S.Wanget S.C.Chen*), '麦冬' (Dry root tubers of *Ophiopogon japonicus (L. f.) Ker Gawl.*)]? Only answer 'Yes' or 'No'.

A:

<think>

Alright, now I will carefully consider the user's question.

The user is asking again whether the composition of "Huganning Capsules" includes *Sedum sarmentosum Bunge*, *Salvia miltiorrhiza Bge.*, *FritiLlaria cirrhosa D.Don*, and *Ophiopogon japonicus*. From the previous conversation history, I have already confirmed the ingredients of Huganning Capsules and provided a clear answer.

This time, the list of ingredients provided by the user does not match the actual ingredients of Huganning Capsules. According to my knowledge base, the main ingredients of Huganning Capsules include 丹参 (Dry rhizomes and roots of *Salvia miltiorrhiza Bge.*), 当归 (Dry roots of *Angelica sinensis (Oliv.) Diels*), 白芍 (Roots of *Cynanchum otophyllum Schneid.*), etc., while *Sedum sarmentosum Bunge*, *FritiLlaria cirrhosa D.Don*, and *Ophiopogon japonicus* are not part of its composition.

Therefore, the correct answer is "No". At the same time, I need to advise the user that if they need detailed information about a medication, they should provide a more accurate name, preferably the full name. This will ensure that accurate information is provided.

Additionally, considering that the user might have some confusion or misunderstanding about the ingredients of traditional Chinese medicine, I can further explain the effects and uses of each medication to help them better understand the composition and usage of traditional Chinese medicine formulas.

</think>

No

Figure 10: The translation of the content in Figure 9.

BIANCANG-QWEN2.5-7B-INSTRUCT

Q: 请问药物：{羚羊角胶囊}的组成成分？
A: 羚羊角胶囊的组成成分包括羚羊角。
# 实际上成分为羚羊角、牛蒡子、淡豆豉、金银花、荆芥、连翘、淡竹叶、桔梗、薄荷素油、甘草

Q: 请问药物：{三七伤药胶囊}的组成成分？
A: 三七伤药胶囊的组成成分包括三七、制草乌、冰片、马钱子粉、南星、木香、乳香（制）、血竭、人工麝香、没药（制）、红花、赤芍、当归、川芎、地黄、泽兰、香附（制）。
# 实际上成分为三七、草乌（蒸）、雪上一枝蒿、接骨木、骨碎补、红花、赤芍、冰片

Q: 请问药物：{健脑安神片}的组成成分？
A: 健脑安神片的组成成分包括黄芪、当归、川芎、白术、茯苓、远志、酸枣仁、龙骨、牡蛎、五味子、柏子仁、合欢皮、夜交藤、珍珠母、地黄、天麻、钩藤、石菖蒲、丹参、红花、川芎、白术、茯苓、远志、酸枣仁、龙骨、牡蛎、五味子、柏子仁、合欢皮、夜交藤、珍珠母、地黄、天麻、钩藤、石菖蒲、丹参、红花、川芎、白术、茯苓、远志、酸枣仁、龙骨、牡蛎、五味子、柏子仁、合欢皮、夜交藤、珍珠母、地黄、天麻、钩藤、石菖蒲、丹参、红花、川芎、白术、茯苓、远志、酸枣仁、龙骨、牡蛎、五味子、柏子仁、合欢皮、夜交藤、珍珠母、地黄、天麻、钩藤、石菖蒲、丹参、红花、川。
# 堆砌常见药材并重复

Figure 11: Examples of BIANCANG-QWEN2.5-7B-INSTRUCT.

BIANCANG-QWEN2.5-7B-INSTRUCT

Q: What are the ingredients of the medication: {Saiga Horns Capsules}?
A: The ingredients of Saiga Horns Capsules include 羚羊角 (Horns of *Saiga Horns Capsules*).
# Actually, the ingredients are 羚羊角 (Horns of *Saiga Horns Capsules*), 牛蒡子 (Dry fruits of *Arctium lappa L.*), 淡豆豉 (Fermented product of the mature seeds of soybeans (*Glycine max (L.) Merr.*)), 金银花 (Dry flowers of *Lonicera japonica Thunb.*), 荆芥 (Dry aerial parts of *Schizonepeta tenuifolia Briq.*), 连翘 (*Forsythia suspensa (Thunb.) Vahl*), 淡竹叶 (Dry stems and leaves of *Lophatherum gracile Brongn.*), 桔梗 (Dry roots of *Platycodon grandiflorus (Jacq.) A.DC.*), 薄荷素油 (Volatile oils made from fresh stems and leaves of *Mentha haplocalyx Briq.*), 甘草 (Dry rhizomes and roots of *Glycyrrhiza uralensis Fisch.*, *Glycyrrhiza inflata Bat.* or *Glycyrrhiza glabra L.*)

Q: What are the ingredients of the medication: {Sanqi Shangyao Capsules}?
A: The ingredients of Sanqi Shangyao Capsules include 三七, 制草乌, 冰片, 马钱子粉, 南星, 木香, 乳香 (制), 血竭, 人工麝香, 没药 (制), 红花, 赤芍, 当归, 川芎, 地黄, 泽兰, 香附 (制).
# Actually, the ingredients are 三七, 草乌 (蒸), 雪上一枝蒿, 接骨木, 骨碎补, 红花, 赤芍, 冰片

Q: What are the ingredients of the medication: {Jiannao Anshen Tablets}?
A: The ingredients of Jiannao Anshen Tablets include 黄芪、当归、川芎、白术、茯苓、远志、酸枣仁、龙骨、牡蛎、五味子、柏子仁、合欢皮、夜交藤、珍珠母、地黄、天麻、钩藤、石菖蒲、丹参、红花、川芎、白术、茯苓、远志、酸枣仁、龙骨、牡蛎、五味子、柏子仁、合欢皮、夜交藤、珍珠母、地黄、天麻、钩藤、石菖蒲、丹参、红花、川芎、白术、茯苓、远志、酸枣仁、龙骨、牡蛎、五味子、柏子仁、合欢皮、夜交藤、珍珠母、地黄、天麻、钩藤、石菖蒲、丹参、红花、川芎、白术、茯苓、远志、酸枣仁、龙骨、牡蛎、五味子、柏子仁、合欢皮、夜交藤、珍珠母、地黄、天麻、钩藤、石菖蒲、丹参、红花、川.
# Piling up incorrect common medicinal materials and repeating.

Figure 12: The translation of the content in Figure 11.

HUATUOGPT2-7B

Q: 请问药物：{心脑健片}的成分是什么？
A: 心脑健片的成分包括：黄芪、葛根、丹参、桂枝、三七、淫羊藿、川芎、何首乌、珍珠、冰片。
# 实际上成分为茶叶

Q: 请问药物：{心脑健片}的成分是[茶叶]吗？
A: 不是的，心脑健片的成分包括黄芪、葛根、丹参、桂枝、三七、淫羊藿、川芎、何首乌、珍珠、冰片，并不包括茶叶。
# 笃定自己的错误

Figure 13: Examples of HUATUOGPT2-7B.

HUATUOGPT2-7B

Q: What are the ingredients of the medication: {Xinnaojian Tablets}?
A: The ingredients of Xinnaojian Tablets include: 黄芪, 葛根, 丹参, 桂枝, 三七, 淫羊藿, 川芎, 何首乌, 珍珠, 冰片.
# Actually, the only ingredient is 茶叶 (Tea leaves).

Q: Whether the only ingredient of medication: {Xinnaojian Tablets} is [茶叶 (Tea leaves)]?
A: No. The ingredients of Xinnaojian Tablets include 黄芪, 葛根, 丹参, 桂枝, 三七, 淫羊藿, 川芎, 何首乌, 珍珠, 冰片, and 茶叶 (Tea leaves) is not included.
# Confirm its incorrectness.

Figure 14: The translation of the examples of HUATUOGPT2-7B in Figure 13.

RETRIEVAL AUGMENTED GENERATION for LLAMA3-CHINESE-8B-INSTRUCT

已知信息：{context}

"你是一个资深的中医与中医药学专家，你具备管理大量药物处方和药材识别的能力；请注意我向你提供了很多《中华人民共和国药典》PDF文件中的内容，但每个PDF文件中包括药方、制备方法、药材特性等内容，请仔细识别各个信息。

现在需要你帮我分析每个中药或中成药的组成成分，只需要为我提供药典中此药品的各个组成成分药材名称即可，无需给出制备方法或用量等详细信息。然后帮我完成以下两个功能：

1.当我给出一个中药名称和一组组成配方表时，给出其正确与否的答案。当基于我提供的药典信息，你认为问题中给出的配方不正确或者不匹配所提供药名时，只需回答"否"，反之，只需回答"是"即可。

2.当我只给出一个中药名称时，请给出你认为此药物正确的配方表，只需要包含成分的药材名称即可，无需具体含量和制备过程。

请务必注意：

有些药物的名称中可能含有类似中医药材但实际上不是药材的名称，切记不要望文生义；

有些药材经常用于制备药物，切记要根据所提供的内容谨慎地判断这些常见药材是否在当前给出的药物中也存在；

请仔细识别药物成分，不要重复给出药材名称。"

请回答以下问题：{question}
"""

Figure 15: The template of RAG for LLAMA3-CHINESE-8B-INSTRUCT.

---

RETRIEVAL AUGMENTED GENERATION for LLaMA3-CHINESE-8B-INSTRUCT

Given the information: {context}

"You are a senior expert in traditional Chinese medicine and pharmacology, with the ability to manage a large number of drug prescriptions and identify medicinal materials. Please note that I have provided you with extensive content from PDF files of the *Pharmacopoeia of the People's Republic of China*, each containing information such as prescriptions, preparation methods, and characteristics of medicinal materials. Please carefully identify each piece of information.

Now, I need you to help me analyze the composition of each traditional Chinese drug or Chinese proprietary medicine. Simply provide me with the names of the constituent medicinal materials listed in the Pharmacopoeia for each drug, without including details such as preparation methods or dosages. Then, assist me in completing the following two tasks:

1. When I provide the name of a traditional Chinese drug and a set of constituent ingredients, give a correct or incorrect answer. Based on the Pharmacopoeia information I have provided, if you determine that the given ingredients in the question are incorrect or do not match the provided drug name, simply respond with "No". Otherwise, reply "Yes".

2. When I only provide the name of a traditional Chinese drug, please give what you believe to be the correct list of ingredients for this medicine. Only include the names of the constituent medicinal materials, without specific quantities or preparation processes.

Please pay close attention to the following:

Some drug names may contain terms that resemble traditional Chinese medicinal materials but are not actual medicinal materials. Do not interpret them literally.

Some medicinal materials are commonly used in drug preparation. Be cautious in determining whether these common materials are present in the currently given medicine based on the provided content.

Carefully identify the drug components and avoid repeating the names of medicinal materials."

Please answer the following question: {question}
""""

Figure 16: The translation of the RAG template in Figure 15.

