# OpenReview forum: "Do "New Snow Tablets" Contain Snow? Large Language Models Over-Rely on Names to Identify Ingredients of Chinese Drugs"
_ICLR.cc/2026/Conference — ICLR 2026 Conference Withdrawn Submission_

### Official Review · Reviewer_AXYi · 2025-10-29

**Soundness:** 2
**Presentation:** 3
**Contribution:** 2
**Rating:** 4
**Confidence:** 5

**Summary:**

This paper evaluates general and specialized LLMs on their ability to identify Traditional Chinese Medicine (TCM) ingredients. The authors systematically identify consistent failure modes, such as literal interpretations and overuse of common herbs. To address these gaps, they apply a standard Retrieval-Augmented Generation (RAG) approach using the Chinese Pharmacopoeia, which is shown to improve ingredient verification accuracy from approximately 50% to 82%.

**Strengths:**

- The paper proposes a robust evaluation framework for the specialized TCM domain. Its dual-task design, assessing both ingredient recall and verification, effectively measures the practical application of domain knowledge.
- The paper excels in its empirical analysis of model failures. It rigorously quantifies specific issues like "common herb overuse" (Fig. 3) and critical reliability flaws like response biases (Fig. 4), providing concrete evidence of the models' knowledge gaps.

**Weaknesses:**

- The paper's core contribution is applying the standard RAG framework to address a known LLM limitation (domain-specific knowledge gaps). The resulting performance improvement is expected and incremental, and the paper fails to articulate any specific novelty in its application of RAG to the TCM domain.
- The experimental evaluation is weak, as it omits comparisons against other prominent, state-of-the-art TCM-specific LLMs (e.g., Zhongjing). Furthermore, the paper fails to benchmark the RAG approach against the crucial alternative of Supervised Fine-Tuning (SFT) using the same knowledge corpus.
- The dataset's construction raises validity concerns. With only 220 herbs, it is too small to ensure robust findings. Additionally, the verification task relies on an unjustified 50% data split and a simple True/False format, which may be insufficient for rigorously testing complex model capabilities.

**Questions:**

The methodology for parsing model outputs for both recall and verification tasks is unclear. How were ingredient lists and True/False judgments reliably extracted from free-text responses? If an automated method (e.g., regex) was used, how was this reconciled with the prompts provided in the appendix, which do not appear to enforce a strict output format? If the evaluation was manual, please address the scalability and reproducibility concerns.

---

### Official Review · Reviewer_d7id · 2025-11-01

**Soundness:** 2
**Presentation:** 2
**Contribution:** 2
**Rating:** 2
**Confidence:** 4

**Summary:**

This paper studies the ability of general-purpose and specialized LLMs to accurately identify ingredients in TCM drugs.
They create a new dataset of 220 TCM formulations from the Pharmacopoeia of the People's Republic of China and evaluate the models on two tasks: direct ingredient inquiry and ingredient list verification.
The study reveals consistent failure patterns in current LLMs, including literal interpretation of drug names, overuse of common herbs, and erratic responses.
To address these shortcomings, the paper proposes RAG-based approach that improves the accuracy and grounding. The work highlights critical weaknesses in the pharmacological knowledge of current LLMs and tries to offer a practical method for improving their reliability in TCM.

**Strengths:**

* The paper addresses an underexplored domain of TCM, a field with unique challenges due to its complex and often ambiguous nomenclature.
The creation of a new, specialized dataset for this task is a valuable contribution as it reveals fundamental deficiencies in the knowledge representation of existing LLMs when applied to specialized domains, and calling the need for an effective, practical solution to enhance their clinical reliability.

* The paper provides a systematic and empirical analysis of the limitations of both general-purpose and TCM-specific LLMs. It moves beyond standard multiple-choice evaluations to probe for deeper understanding through direct inquiry and verification tasks, which effectively exposes the models' failure modes. The authors' analysis of error patterns—such as literal interpretation and common herb overuse—is insightful and well-supported by the evidence presented. The proposed RAG-based solution is shown to be highly effective, substantially improving performance across all tested models.

**Weaknesses:**

### Insufficient Scope and Depth of the Proposed Solution's Evaluation
The paper's primary methodological contribution, a RAG-based approach, is evaluated on only a single general-purpose model (LLAMA3). This is a significant limitation, as it fails to establish the generalizability of the solution. As multiple reviewers noted, it leaves unanswered to what extent other models, particularly the TCM-specific ones, would benefit. Given that RAG is a well-established technique, such a narrow evaluation is insufficient to support the paper's claims.

&nbsp;

### Failure to Analyze the Limitations of the RAG Approach
The paper presents RAG as a solution but offers no investigation into its own failure modes or limitations within this domain. A thorough contribution would require an error analysis of the RAG-enabled model to understand what types of errors persist. The work does not clarify whether remaining issues stem from faulty retrieval, the model's inability to reason over correct context, or inherent ambiguities in the source material that RAG cannot resolve.

&nbsp;

### Superficial Diagnosis of the Core Problem
The paper does not adequately explore the fundamental reasons why even domain-specific LLMs fail. While it demonstrates that RAG—a knowledge-injection method—is effective, this suggests the problem may simply be a knowledge deficit. The paper lacks a deep analysis of the training data and objectives of models like HuatuoGPT to definitively prove that the failure is due to a more complex issue like a "task alignment" gap rather than insufficient exposure to the right kind of factual data.

&nbsp;

### Limited Scale of the Evaluation Dataset
The core claims of systematic model failure are based on a dataset of 220 drugs. While this was sufficient to reveal certain error patterns, this small scale undermines the claim of a "thorough" or "systematic" evaluation, especially when the novelty of the technical approach is limited.

**Questions:**

* Could you please provide more comprehensive results on how the proposed RAG approach work on other baseline models?
* Fundamental challenges beyond knowledge retrieval: Your results show that providing factual context via RAG dramatically improves performance. Does this imply that the identified failures are primarily an issue of knowledge memorization rather than a deeper flaw in reasoning? What evidence can you provide that there are fundamental challenges in this domain that cannot be solved simply by retrieving and presenting the correct information to the model? Also, you note that both general and TCM-specific LLMs perform poorly. However, their failures likely stem from different reasons, in such specialized domains like TCM. Could you provide a more detailed comparative analysis?
* Exploring the failure modes of RAG: What are the limitations of your proposed RAG solution? Can you provide an analysis of the cases where the LLAMA3 + RAG model still fails? What do these specific failures tell us about the remaining challenges in automating ingredient identification in TCM?

---

### Official Review · Reviewer_66kQ · 2025-11-02

**Soundness:** 1
**Presentation:** 2
**Contribution:** 1
**Rating:** 2
**Confidence:** 3

**Summary:**

This paper investigates whether large language models can accurately identify the ingredients of Traditional Chinese Medicine (TCM) drugs. The authors evaluate models on two tasks: (1) directly listing ingredients for 220 Chinese proprietary medicines, and (2) verifying whether a given ingredient list matches a drug name. Their analysis shows consistent failure patterns: models rely heavily on literal interpretation of drug names, overuse common herbs, and often behave inconsistently, indicating shallow understanding rather than true pharmacological knowledge. To address this, the authors introduce a retrieval-augmented generation (RAG) approach using Pharmacopoeia-based lookup, improving ingredient verification accuracy from ~50% to ~82%. The work highlights serious reliability gaps in current TCM-oriented LLMs and proposes a practical inference-time remedy.

**Strengths:**

- This work constructs a dataset for evaluating LLM's ability to identify drug ingredients in traditional Chinese medicine, which is an unexplored area.

**Weaknesses:**

- I don't think this work is significant enough for ICLR venue. The contribution is very limited. First, the proposed dataset is relatively small, only from 220 traditional Chinese medicine drugs. Second, the evaluation is very limited. For direct ingredient inquiry task, only BIANCANG-QWEN2.5-7B-INSTRUCT's results are reported, and no RAG result is reported. For ingredient list verification, RAG result is only reported for LLAMA3-CHINESE-8B-INSTRUCT, but no other models.
- The metrics are not discussed, but only revealed later in the figures and tables in section 4. I would suggest using a unified metric for direct ingredient inquiry task rather than reporting for each ingredient. In this way, you can compare different models more easily.
- Some models (e.g. ChiMedGPT) mentioned in the related work are not evaluated.

**Questions:**

- Why ChiMedGPT mentioned in Section 2.1 is not tested? Also, you may want to remove the parentheses for "Yuanhe Tian, 2023" on line 130.
- What are the total numbers of questions for direct ingredient inquiry and ingredient list verification?
- Which retriever do you use for RAG?
- In Section 4.2, why do you need to ask 220 question in a random order?

---

### Official Review · Reviewer_4ZXm · 2025-11-04

**Soundness:** 3
**Presentation:** 2
**Contribution:** 1
**Rating:** 2
**Confidence:** 4

**Summary:**

The authors investigate the ability of LLMs, both general-purpose and fine-tuned for the purpose, to identify the ingredients of traditional Chinese medicines (TCMs) and identify that all models tested perform poorly without RAG.

**Strengths:**

- The paper clearly demonstrates that existing language models are incapable of reliably identifying the ingredients of TCMs.
- The paper further shows that the LLMs studied are biased towards particular answers (e.g. always predicting that a given ingredient is present, or the reverse) and prevent observational evidence that LLMs may be heavily biased by literal interpretations of medicine names.

**Weaknesses:**

- The paper has limited novelty in terms of generalizable principles relevant to the wider ML community. The key findings are that existing models (whose training data emphasizes standard medical practice rather than TCM) fail at ingredient identification, and that RAG, as would be expected, benefits the task.
- The paper may be more appropriate to a medical application-focused venue.

**Questions:**

Further discussion of the failure of the TCM-specific LLMs may be helpful. Is this due to a limitation of the training pipeline and datasets? Do the TCM-specific LLMs outperform generic LLMs in the presence of RAG?

The results would be strengthened with the incorporation of a confidence measurement (e.g. derived from the LLM's output distribution) and computation of AUC-ROC.

The writing needs to be improved; constructions such as "the performance of a model in real applications is unaware" [170] should be made more clear, and informal language such as "the RAG is adding a prosthetic brain for the language model..." [444] should be limited. Certain statements such as "...which demonstrates that the only weakness of DeepSeek-R1 is the lack of knowledge" [435] as a consequence of the model's reasoning nature do not appear justified, and empirical evidence is needed. For this to be true, the model should perform perfectly with RAG. Comparison to additional reasoning models would be needed to identify whether any improved performance of the model is due to reasoning or alternative factors, e.g. greater prevalence of TCM data in training.

---

### Note · Authors · 2026-01-26

I have read and agree with the venue's withdrawal policy on behalf of myself and my co-authors.

---

### Meta-Review · Area_Chair_y2tk · 2025-12-31

**Summary:**

This paper investigates how Large Language Models, both general-purpose and TCM-specialized, identify ingredients in Traditional Chinese Medicine (TCM) formulations. The authors create a dataset of 220 TCM drugs from the Chinese Pharmacopoeia and evaluate models on two tasks: direct ingredient inquiry and ingredient list verification. Their analysis reveals consistent failure patterns across all models: literal interpretation of drug names, overuse of common herbs regardless of relevance, and erratic behavior when faced with unfamiliar formulations. The authors also propose an Retrieval-Augmented Generation approach that significantly improves verification accuracy. The reviewers acknowledged the paper's empirical value in exposing critical gaps in LLM knowledge representation for specialized domains. However, they raised fundamental concerns about the novelty of the technical contribution, limited evaluation scope, dataset size constraints, and methodological transparency that collectively undermine the paper's suitability for ICLR.

**Reviewer Concerns:**

The authors didn't make rebuttal.

Outstanding concerns:
 - Novelty of technical approach (Reviewers 4ZXm, AXYi, and d7id): Reviewers all questioned whether applying standard RAG to this domain constitutes a significant contribution for ICLR. The paper lacks evidence of novel RAG adaptations specific to TCM challenges beyond standard implementation.
 - Limited dataset coverage: All reviewers noted the small dataset size (220 formulations) as insufficient for systematic claims.
 - Missing model evaluations (Reviewers 66kQ and AXYi): Reviewers noted critical omissions of prominent TCM-specific models mentioned in related work (ChiMedGPT, Zhongjing) from the evaluation.

**Reviewer Scores:**

- Reviewer 4ZXm (original score: 2): Would maintain at 2.
 - Reviewer 66kQ (original score: 2): Would maintain at 2.
 - Reviewer d7id (original score: 2): Would maintain at 2.
 - Reviewer AXYi (original score: 4): Would maintain at 4.

---

### Decision · Program_Chairs · 2026-01-26

Reject